# Pro-Inflammatory Cytokines Enhanced In Vitro Cytotoxic Activity of *Clostridioides difficile* Toxin B in Enteric Glial Cells: The Achilles Heel of *Clostridioides difficile* Infection?

**DOI:** 10.3390/ijms25020958

**Published:** 2024-01-12

**Authors:** Katia Fettucciari, Andrea Spaterna, Pierfrancesco Marconi, Gabrio Bassotti

**Affiliations:** 1Biosciences & Medical Embryology Section, Department of Medicine and Surgery, University of Perugia, 06132 Perugia, Italy; pierfrancesco.marconi@outlook.it; 2School of Biosciences and Veterinary Medicine, University of Camerino, 62024 Macerata, Italy; 3Gastroenterology, Hepatology & Digestive Endoscopy Section, Department of Medicine and Surgery, University of Perugia, 06132 Perugia, Italy; gabrio.bassotti@unipg.it; 4Gastroenterology & Hepatology Unit, Santa Maria Della Misericordia Hospital, 06129 Perugia, Italy

**Keywords:** *Clostridioides difficile*, *Clostridioides difficile* toxin B, apoptosis, necrosis, pro-inflammatory cytokines, tumor necrosis factor alpha, interferon gamma, inflammation, cytotoxicity, cell death

## Abstract

Bacterial infections are characterized by an inflammatory response, which is essential for infection containment but is also responsible for negative effects on the host. The pathogen itself may have evolved molecular mechanisms to antagonize the antimicrobial effects of an inflammatory response and to enhance its pathogenicity using inflammatory response mediators, such as cytokines. *Clostridioides difficile* (*C. difficile*) infection (CDI) causes gastrointestinal diseases with markedly increasing global incidence and mortality rates. The main *C. difficile* virulence factors, toxin A and B (TcdA/TcdB), cause cytopathic/cytotoxic effects and inflammation. We previously demonstrated that TcdB induces enteric glial cell (EGC) apoptosis, which is enhanced by the pro-inflammatory cytokine tumor necrosis factor alpha plus interferon gamma (CKs). However, it is unknown whether CKs-enhanced TcdB cytotoxicity (apoptosis/necrosis) is affected by the timing of the appearance of the CKs. Thus, we simulated in vitro, in our experimental model with TcdB and EGCs, three main situations of possible interactions between TcdB and the timing of CK stimulation: before TcdB infection, concomitantly with infection, or at different times after infection and persisting over time. In these experimental conditions, which all represent situations of possible interactions between *C. difficile* and the timing of CK stimulation, we evaluated apoptosis, necrosis, and cell cycle phases. The CKs, in all of these conditions, enhanced TcdB cytotoxicity, which from apoptosis became necrosis when CK stimulation persisted over time, and was most relevant after 48 h of TcdB:EGCs interaction. Particularly, the enhancement of apoptosis by CKs was dependent on the TcdB dose and in a less relevant manner on the CK stimulation time, while the enhancement of necrosis occurred always independently of the TcdB dose and CK stimulation time. However, since in all conditions stimulation with CKs strongly enhanced the TcdB cytotoxicity, it always had a negative impact on *C. difficile* pathogenicity. This study might have important implications for the treatment of CDI.

## 1. Introduction

Bacterial infections are characterized by a strong inflammatory response considered essential for the containment of the infection and the replication of the microorganisms [1,2,3,4]. Of course, the negative effects of this inflammatory host response, both locally and systemically, have been studied in depth [5]. However, recent studies suggest that pathogens may have evolved molecular mechanisms to antagonize the negative effects of the inflammatory response on their molecular strategy of infection [6,7,8]. The question remains open whether bacteria may have also evolved mechanisms to enhance their pathogenicity using mediators of the inflammatory response, such as cytokines.

*Clostridioides difficile* (*C. difficile*) [9,10] is a Gram-positive, anaerobic, spore-forming bacterium that is responsible for about of 15–25% of all opportunistic gastrointestinal infections and >90% of mortality resulting from these infections [11,12,13]. A *C. difficile* infection (CDI) can appear in a pauci-symptomatic form with mild transient diarrhoea, or in severe symptomatic forms featuring pseudomembranous colitis, toxic megacolon, colonic perforation, and death [14,15,16]. *C. difficile* produces toxin A (TcdA), toxin B (TcdB), and a binary toxin (CTD) [17,18,19]. The pathogenicity of this bacterium is mainly due to its two exotoxins (Tcds), TcdA and TcdB, capable of inducing all of the clinical manifestations associated with CDI [17,18,19] by provoking disruption of the colonic epithelia barrier, activation of immune cells, and stimulation of the release of pro-inflammatory cytokines and chemokines [20,21]. TcdB is about 1000 times more potent than TcdA in terms of cytotoxic activity in several cell lines [22,23,24], and it plays a major role in overall CDI pathogenesis because it is capable of causing the full spectrum of disease [19,25,26,27]. 

Tcds, by glucosylation of Rho-GTPases at the catalytic site, inhibit their functions, causing several glucosylation-dependent effects in vitro and in vivo (Figure 1) [17,18,28,29,30,31,32]; Tcds also induce glucosyltransferase-independent effects both in vitro and in vivo (Figure 1) [17,18,30,31,32].

The importance of glucosylation-independent necrosis during CDI has remained an outstanding question [32]. Recently, it has been demonstrated, in a mouse model of CDI, that epithelial damage arises by a glucosylation-independent way that does not require/elicit the recruitment of immune cells [32,33,34].

Tcds, by causing damage to the epithelium, stimulate the host inflammatory and immune responses by which the immune system attempts to constrain epithelial damage and the dissemination of intestinal bacteria into the circulation. However, an excessively strong and prolonged inflammatory response during CDI can be damaging to the host and contribute to disease pathology by increasing the severity of tissue damage and the probability of a lethal disease outcome [2,20,33,34,35]. Certainly, relapses increase the degree and duration of the inflammatory response [36].

We have previously demonstrated in vitro that apoptosis induced by TcdB is strongly enhanced by the presence of pro-inflammatory cytokines, tumor necrosis factor alpha (TNF-α) plus interferon gamma (IFN-γ) (CKs) [28,37]. In this cytotoxic synergism, the three main apoptotic pathways already activated by TcdB alone, mediated by calpains, caspases, and cathepsin B, are strongly enhanced [37].

In the present study, using our in vitro TcdB infection model of enteric glial cells (EGCs), cells central to the regulation of intestinal pathophysiology [38,39,40,41], we investigated how *C. difficile* cytotoxicity is affected in relation to the timing of stimulation with CKs, evaluating the cytotoxic activity as both apoptosis and necrosis. To this purpose, we simulated in vitro three important situations of possible interactions between *C. difficile* and the timing of pro-inflammatory CKs stimulation, considered as the response of the host. CDI in a host: (I) with an inflammatory state preceding the infection, (II) with an inflammatory response of the host that begins at the start of the infection, or (III) with an inflammatory response that becomes significant over time. 

The results obtained demonstrate that stimulation with CKs both before but also after a relatively longer time of infection with TcdB enhances the cytotoxic activity (apoptosis and necrosis) of TcdB. Furthermore, even where TcdB alone does not cause cytotoxic effects, these can be induced by pro-inflammatory CKs.

Furthermore, this offers a pre-clinical model study to better understand the pathophysiology of CDI in the perspective of a new approach for the treatment of CDI. 

## 2. Results

Previously, we demonstrated in vitro that the apoptotic effect of TcdB on EGCs is strongly enhanced by pro-inflammatory CKs [28,37].

In this study, we investigated in vitro whether the timing of stimulation with CKs enhanced the pathogenicity of TcdB, evaluated as both apoptosis and necrosis. Thus, we simulated in vitro the three main situations of possible interactions between *C. difficile* and the timing of pro-inflammatory CK stimulation, considered as the response of the host:

First situation: CDI in a host with an inflammatory state preceding the infection, ranging from subjects with a low-grade inflammatory state to subjects with a strong-grade inflammatory state as occurs in Inflammatory Bowel Disease. This situation is simulated by pre-treating EGCs for 18 h or 2 h with CKs before treatment with TcdB (scheme A and B in Figure 2);

Second situation: CDI with an inflammatory response of the host that begins at the start of the infection. This situation is simulated by treating EGCs with TcdB and concomitantly or after 1.5 h with CKs (scheme C and D in Figure 2);

Third situation: CDI in subjects in whom the inflammatory response becomes significant over time. This situation is simulated by: (I) treating EGCs with TcdB and after 48 h with CKs for 24 h or 48 h (scheme E in Figure 2), and (II) treating EGCs with TcdB and after 1.5 h with CKs for 24 h, 48 h, or 72 h (scheme F and G in Figure 2).

For all of the schemes, at the time indicated the following parameters were evaluated: (a) cell death by apoptosis, measuring the percentage of apoptotic cells (the hypodiploid DNA content) by flow cytometry, (b) cell death by necrosis, evaluating cell viability and the live cell number by erythrosine B staining, and (c) cell-cycle distribution, evaluating the DNA content by flow cytometry analysis.

All data are expressed as the mean ± SD of six experiments. Statistical analysis was conducted using GraphPad Prism 9.0.0 software. The Shapiro–Wilk test was applied for testing the data normality distribution. For evaluating the significance of the differences between the groups, *p*-values were calculated using ordinary one-way ANOVA followed by Tukey’s post hoc test because the data were normally distributed. *p*-values = or less than 0.05 were defined as statistically significant.

The various schemes all represent situations of possible interactions between *C. difficile* and the timing of CK stimulation; therefore, there is no advantage or benefit of one scheme over the other, but the study of all of them is important to understand the role of the inflammatory response in the pathogenesis of CDI.

In the schemes chosen to evaluate the effects of cytotoxic synergism between TcdB and the timing of the appearance of CKs, attempting to simulate what could happen in vivo, we continued the observation time of the effects after 24 h. For a more detailed analysis, we have chosen the specific time intervals from 1.5 h to 24 h, 48 h, 72 h, or 96 h from infection, because cell death, and especially apoptosis (but also necrosis), could occur after a longer time from triggering.

### 2.1. Effect of TcdB on EGCs Already in an Environment with Pro-Inflammatory CKs

#### 2.1.1. Pro-Inflammatory CKs at 18 h before TcdB

To evaluate the TcdB activity on EGCs that have already interacted with cytokines, i.e., an environment characterized by a pro-inflammatory state, we pre-treated EGCs for 18 h (−18 h) with CKs before infection with TcdB at doses of 0.1 ng/mL, 1 ng/mL, and 10 ng/mL (time 0), and then the effects were evaluated at 24 h (scheme A in Figure 2). 

The results demonstrated that TcdB alone induces apoptosis in EGCs in a dose-dependent manner (Figure 3A). Pre-treatment of EGCs for 18 h with CKs induced a significant increase in apoptosis with respect to TcdB alone at the doses of 0.1 ng/mL and 1 ng/mL, while there was no significant increase in apoptosis with a dose of 10 ng/mL (Figure 3A). In fact, we found a 2.4- and 1.4-fold increase of apoptosis, respectively, with 0.1 ng/mL and 1 ng/mL TcdB, while we did not find a significant increase in apoptosis with TcdB at 10 ng/mL (Figure 3A).

The results of the cell viability assay, measured by erythrosine B staining of the dead cells, indicative of cell death by necrosis, showed that TcdB alone induces cell death by necrosis (Figure 3B,C). In fact, we observed: (a) 16.8%, 20.8%, and 20.8% erythrosine B-positive cells in EGCs treated, respectively, with TcdB alone at 0.1 ng/mL, 1 ng/mL, and 10 ng/mL (Figure 3B); and (b) a strong reduction in the number of live cells (Figure 3C) at all concentrations of TcdB used with respect to control EGCs (Figure 3C). Pre-treatment of EGCs for 18 h with CKs induced a significant increase in erythrosine B-positive cells in EGCs treated with TcdB+CKs with respect to EGCs treated with TcdB alone (Figure 3B). In fact, we found a 1.85-, 1.4- and 1.37-fold increase of erythrosine B-positive cells, respectively, with TcdB at 0.1 ng/mL, 1 ng/mL, and 10 ng/mL after pre-treatment of EGCs for 18 h with CKs (Figure 3B). Pre-treatment of EGCs for 18 h with CKs did not further reduce the number of live cells with respect to that of EGCs treated with TcdB alone (Figure 3C).

Pre-treatment of EGCs for 18 h with CKs alone had not significant cytotoxicity in EGCs (Figure 3). 

TcdB alone induces a cell cycle arrest in G2/M; indeed, we found an increase of cells in the G2/M phase with the disappearance of the S phase (Figure 3D). The same phenomenon occurred after treatment with TcdB in cells pre-treated for 18 h with CKs (Figure 3D). Moreover, pre-treatment for 18 h with CKs (Figure 3D) also induced a significant increase in the percentage of cells in G0/G1 and G2/M phase in EGCs treated with TcdB at 10 ng/mL (Figure 3D).

#### 2.1.2. Pro-Inflammatory CKs at 2 h before TcdB

To evaluate the TcdB activity on EGCs that have recently interacted with cytokines, we pre-treated EGCs with CKs for 2 h (−2 h) before infection with TcdB at doses of 0.1 ng/mL, 1 ng/mL, and 10 ng/mL, and then the effects were evaluated at 24 h (scheme B in Figure 2).

The results demonstrated that pre-treatment of EGCs with CKs for 2 h (Figure 4A) led to effects similar to those described with pre-treatment with CKs for 18 h (Figure 3A). In fact, there was a significant increase in apoptosis with TcdB at 0.1 ng/mL and 1 ng/mL, while there was no significant increase in apoptosis with TcdB at 10 ng/mL (Figure 4A). Particularly, we found a 1.73- and 1.3-fold increase of apoptosis, respectively, with 0.1 ng/mL and 1 ng/mL TcdB, while we did not find a significant increase in apoptosis with TcdB at 10 ng/mL (Figure 4A).

The results of the cell viability assay by erythrosine B staining of dead cells showed that pre-treatment of EGCs for 2 h with CKs induced a significant increase of erythrosine B-positive cells in EGCs treated with TcdB+CKs with respect to EGCs treated with TcdB alone (Figure 4B). In fact, we found a 1.51-, 1.56-, and 1.53-fold increase of erythrosine B-positive cells, respectively, with TcdB at 0.1 ng/mL, 1 ng/mL, and 10 ng/mL (Figure 4B). Pre-treatment of EGCs for 2 h with CKs did not further reduce the number of live cells with respect to treatment with TcdB alone (Figure 4C).

Pre-treatment of EGCs for 2 h with CKs alone did not have significant cytotoxicity in EGCs (Figure 4). 

TcdB alone induces cell cycle arrest in G2/M, and indeed, we found an increase of cells in the G2/M phase, with the disappearance of the S phase (Figure 4D). The same phenomenon occurred after treatment with TcdB in cells pre-treated with CKs for 2 h (Figure 4D). Moreover, pre-treatment with CKs for 2 h also induced a significant increase in the percentage of cells in the G0/G1 and G2/M phases in EGCs treated with TcdB at 10 ng/mL (Figure 4D).

### 2.2. Effect of TcdB on EGCs Concurrently with Pro-Inflammatory CKs

The production and secretion of TcdB by *C. difficile* in vivo could be accompanied by the production of cytokines because of the rapid inflammatory state that can be established. Therefore, to simulate the conditions reported above (i.e., an environment in which a pre-existing inflammatory state does not exist) the EGCs were incubated with different doses of TcdB (0.1 ng/mL, 1 ng/mL, or 10 ng/mL) and treated simultaneously (0 h) with CKs, and then the effects were evaluated at 24 h (scheme C in Figure 2).

Treatment of EGCs with CKs concomitantly with TcdB induced a significant increase in apoptosis with 0.1 ng/mL and 1 ng/mL, while there was no significant increase in apoptosis with 10 ng/mL (Figure 5A). In fact, we found a 1.73- and 1.3-fold increase of apoptosis, respectively, with 0.1 ng/mL and 1 ng/mL (Figure 5A), while we did not find a significant increase in apoptosis at 10 ng/mL (Figure 5A).

Treatment of EGCs with CKs concomitantly with TcdB induced a significant increase of erythrosine B-positive cells in EGCs treated with TcdB+CKs with respect to EGCs treated with TcdB alone (Figure 5B). In fact, we found a 1.61-, 1.5- and 1.59-fold increase of erythrosine B-positive cells, respectively, with TcdB at 0.1 ng/mL, 1 ng/mL, and 10 ng/mL (Figure 5B). Regarding the number of live cells (Figure 5C), treatment of EGCs with CKs concomitantly with TcdB did not further reduce the number of live cells with respect to that of EGCs treated with TcdB alone (Figure 5C). 

CKs alone did not have significant cytotoxicity in EGCs (Figure 5). 

Treatment with CKs concomitantly with TcdB induces cell cycle arrest in G2/M, and indeed, we found an increase of cells in the G2/M phase with disappearance of the S phase as demonstrated for TcdB alone (Figure 5D). However, treatment with CKs concomitantly with TcdB also induced a significant increase in the percentage of cells in the G0/G1 and G2/M phases in EGCs treated with TcdB at 10 ng/mL (Figure 5D).

### 2.3. Effect of Pro-Inflammatory CKs Given at a Brief Time (+1.5 h) from TcdB Infection

It is possible that in vivo, shortly after the onset of the CDI, after a brief period of latency the production of cytokines begins. Therefore, to simulate the conditions reported above (i.e., an environment in which a pre-existing inflammatory state does not exist) the EGCs were treated with TcdB at 0.1 ng/mL, 1 ng/mL, or 10 ng/mL, and after 1.5 h (+1.5 h) they were treated with CKs. Then, the effects were evaluated at 24 h (scheme D in Figure 2).

Treatment with CKs 1.5 h after TcdB induces effects similar to those obtained when CKs are added simultaneously or before TcdB (Figure 3A, Figure 4A and Figure 5A). In fact, CKs added 1.5 h after TcdB induced a significant apoptosis increase at 0.1 ng/mL and 1 ng/mL, while it did not significantly increase apoptosis with 10 ng/mL (Figure 6A). We found a 1.97- and 1.31-fold increase of apoptosis at 0.1 and 1 ng/mL (Figure 6A), while there was no significant increase in apoptosis at 10 ng/mL (Figure 6A).

Treatment of EGCs with CKs 1.5 h after TcdB induced a significant increase of erythrosine B-positive cells in EGCs treated with TcdB (Figure 6B). In fact, we found, a 1.76-, 1.5-, and 1.62-fold increase of erythrosine B-positive cells, respectively, with 0.1 ng/mL, 1 ng/mL, and 10 ng/mL (Figure 6B). Treatment of EGC with CKs 1.5 h after TcdB did not further reduce the number of live cells with respect to that of EGCs treated with TcdB alone (Figure 6C).

Also, in these experimental conditions, CKs alone did not display significant cytotoxicity in EGCs (Figure 6). 

Treatment with CKs 1.5 h after TcdB induces cell cycle arrest in G2/M, and indeed, we found an increase of cells in the G2/M phase with the disappearance of the S phase as demonstrated for TcdB alone (Figure 6D). However, treatment of EGCs with CKs 1.5 h after TcdB in EGCs treated with 10 ng/mL also induced a significant increase in the percentage of cells in the G0/G1 and G2/M phases (Figure 6D).

### 2.4. Kinetics of Effect of CKs Given at a Longer Time (+48 h) from TcdB Infection

It is possible in vivo that *C. difficile* infections begin slowly with relatively low TcdB secretion and that only after 24 h to 48 h is there a significant pro-inflammatory cytokine response that increases as the *C. difficile* infection worsens. Therefore, to simulate this condition, EGCs were incubated with TcdB (0.1 ng/mL, 1 ng/mL, or 10 ng/mL) and after 48 h (+48 h) were stimulated with CKs for 24 h or 48 h and the effects were evaluated after 24 h (+72 h from time 0) or 48 h (+96 h from time 0) (scheme E in Figure 2).

The results obtained showed that the percentage of apoptosis with TcdB alone at 0.1 ng/mL, 1 ng/mL, or 10 ng/mL after 24 h (+72 h from time 0) was, respectively, 19.8%, 27.4%, and 24.2% (Figure 7A); and after 48 h (+96 h from time 0) was, respectively, 22.6%, 18.6%, and 20.3% (Figure 7B). Treatment with CKs for 24 h (+72 h from time 0) or 48 h (+96 h from time 0) after 48 h of TcdB induced an enhancement of apoptosis (Figure 7A,B). In fact, the percentage of apoptosis after 24 h (+72 h from time 0) was, respectively, 34.6%, 57.8%, and 49.18% with TcdB at 0.1 ng/mL, 1 ng/mL, and 10 ng/mL (Figure 7A); and after 48 h (+96 h from time 0) it was increased, respectively, to 43.8%, 52.6%, and 44.7% with TcdB at 0.1 ng/mL, 1 ng/mL, and 10 ng/mL (Figure 7B).

Regarding the results of the loss of cell viability, the percentage of erythrosine B-positive cells in EGCs treated with TcdB alone after 24 h (+72 h from time 0) was, respectively, 18.76%, 30.72%, and 31.27% with TcdB at 0.1 ng/mL, 1 ng/mL, and 10 ng/mL (Figure 7C), and it did not change significantly after treatment with TcdB alone for 48 h (+96 h from time 0); indeed, it was, respectively, 20.35%, 28.60%, and 31.0% with TcdB at 0.1 ng/mL, 1 ng/mL, and 10 ng/mL (Figure 7D). Treatment with CKs for 24 h (+72 h from time 0) or 48 h (+96 h from time 0) after 48 h of TcdB induced an enhancement of necrosis (an increase of the percentage of erythrosine B-positive cells) (Figure 7C,D) In fact, the percentage of erythrosine B-positive cells after 24 h (+72 h from time 0) was, respectively, 24.63%, 43.03%, and 47.30% with TcdB at 0.1 ng/mL, 1 ng/mL, and 10 ng/mL (Figure 7C); and after 48 h (+96 h from time 0) it was increased, respectively, to 33.6%, 52.0%, and 51.2% with TcdB at 0.1 ng/mL, 1 ng/mL, and 10 ng/mL (Figure 7D).

The results of the cell viability assay by erythrosine B staining of dead cells showed that the number of live cells was decreased after 24 h (+72 h from time 0) and further decreased after 48 h (+96 h from time 0) (Figure 7E,F).

Analysis of the cell cycle showed that treatment with TcdB alone for 24 h (+72 h from time 0) or 48 h (+96 h from time 0) induced a strong cell cycle arrest in G2/M of EGCs, and indeed, we found an increase of cells in the G2/M phase with the disappearance of the S phase (Figure 7G,H) and that treatment with CKs for 24 h (+72 h from time 0) or 48 h (+96 h from time 0) did not further increase EGC cell cycle arrest in G0/G1 and G2/M (Figure 7G,H). Therefore, also at these times, the cell cycle arrest in G2/M was maintained and treatment with CKs for 24 h and 48 h did not further affect cell cycle arrest. 

Therefore, CKs, when added a longer time from TcdB infection, strongly enhanced the cytotoxic activity of TcdB, both as apoptosis and necrosis (Figure 7A–F), and the cytotoxic activity (apoptosis and necrosis) of TcdB+CKs with all doses of TcdB increased with time and was greater when EGCs that have encountered TcdB were stimulated with CKs for 48 h (+96 h from time 0, scheme E in Figure 2) (Figure 7B,D,F).

Overall, these data showed that the enhancement of the cytotoxic activity of TcdB by CKs also occurs at 48 h after TcdB infection and also has a progressive action, leading to the death of a high proportion of cells.

### 2.5. Kinetics of the Effect of CKs Given at a Brief Time [Close] (+1.5 h) from TcdB Infection

To verify whether this progressive increase in cytotoxicity is due to the TcdB+CKs cytotoxic synergism and therefore is also present when the CKs are given close to the TcdB, we performed two series of experiments.

In the first series, we performed time-dependent experiments up to 48 h with all of the doses of TcdB used and the addition of CKs 1.5 h after TcdB. To this purpose, EGCs were incubated with TcdB at 0.1 ng/mL, 1 ng/mL, or 10 ng/mL, and after 1.5 h (+1.5 h) they were treated with CKs, and then the effects were evaluated at 24 h and 48 h (scheme F in Figure 2).

Concerning apoptosis, the results demonstrated that the percentage of apoptosis induced by TcdB alone at 48 h did not significantly change with respect to that of 24 h, and it decreased with the higher doses of TcdB (Figure 6E). In fact, we found 13.8% at 24 h and 16.05% at 48 h with 0.1 ng/mL; 34.89% at 24 h and 31.38% at 48 h with 1 ng/mL; and 36.77% at 24 h and 24.77% at 48 h at 10 ng/mL (Figure 6E). Treatment with CKs 1.5 h after TcdB increased the percentage of apoptosis at 48 h with all doses of TcdB (Figure 6E), although the strongest increase of apoptosis was found with TcdB at 0.1 ng/mL (Figure 6E).

The results of necrosis obtained by the analysis of cell viability loss demonstrated that at 48 h, the percentage of erythrosine B-positive cells induced by TcdB alone at 0.1 ng/mL did not significantly change with respect to that of 24 h, while it strongly increased at 1 ng/mL and 10 ng/mL (Figure 6F). Treatment with CKs 1.5 h after TcdB strongly increased the percentage of erythrosine B-positive cells at 48 h with all of the doses of TcdB used (Figure 6F).

The results of the live cell number showed that at 48 h, the number of live cells with TcdB alone a 0.1 ng/mL was reduced by about 1.29-fold (Figure 6G) with respect to that of 24h (Figure 6C). Also after treatment with TcdB alone at 1 ng/mL and 10 ng/mL, the number of live cells was significantly decreased at 48 h (Figure 6G) with respect to that of 24 h (Figure 6C). In fact, we found a decrease of live cell number of about 1.23-fold with TcdB alone at 1 ng/mL and 2.13-fold with TcdB at 10 ng/mL (Figure 6G) with respect to that of 24 h (Figure 6C). Further, treatment with CKs 1.5 h after TcdB with all of the doses of TcdB used (0.1 ng/mL, 1 ng/mL, and 10 ng/mL) strongly decreased the number of live cells at 48 h (Figure 6G) with respect both to that of TcdB alone (Figure 6G) and to that of TcdB+CKs at 24 h (Figure 6C). 

Analysis of the cell cycle showed that after treatment with TcdB alone at all doses used there was a strong cell cycle arrest in G2/M of EGCs with the disappearance of the S phase both at 24 h (Figure 6D) and 48 h (Figure 6H). Treatment with CKs 1.5 h after TcdB did not further increase EGC cell cycle arrest in G2/M (Figure 6D,H). Therefore, at 48 h the cell cycle arrest in G2/M was also maintained and treatment with CKs did not further affect the cell cycle arrest.

In the second series, we performed time-dependent experiments, always with the addition of CKs 1.5 h after TcdB but prolonging the observation time up to 72 h, choosing only the dose of TcdB of 0.1 ng/mL because this is the dose that induces the greatest enhancement of the cytotoxicity (apoptosis and necrosis) (scheme G in Figure 2). EGCs were incubated with TcdB at 0.1 ng/mL and after 1.5 h treated with CKs, and then the effects were analysed at 24 h, 48 h, and 72 h (scheme G in Figure 2).

Regarding analysis of apoptosis, the results demonstrated that the percentage of apoptosis induced by TcdB (0.1 ng/mL) alone at 48 h did not significantly change with respect to that of 24 h, while it was increased at 72 h (Figure 8A–C). In fact, we found 12.0% at 24 h, 14.3% at 48 h, and 19.6% at 72 h (Figure 8A–C). Regarding apoptosis induced by treatment with CKs 1.5 h after TcdB (0.1 ng/mL), the percentage of apoptosis increased progressively up to 72 h, increasing at 48 h by about 2.96-fold with respect to the value found with TcdB alone at 48 h and 3.51-fold with respect to the value found with TcdB alone at 24 h; and at 72 h by 3.01-fold with respect to the value found with TcdB alone at 72 h and by 4.9-fold with respect to the value found with TcdB alone at 24 h (Figure 8A–C). In fact, we found about 25.3% apoptosis at 24 h, 42.3% at 48 h, and 59.1% at 72 h (Figure 8A–C).

The results of necrosis from the analysis of cell viability loss demonstrated that the percentage of erythrosine B-positive cells induced by TcdB (0.1 ng/mL) alone at 48 h did not significantly change with respect to that of 24 h, but increased at 72 h (Figure 8D–F). In fact, we found 13.4% at 24 h, 15.2% at 48 h, and 22.3% at 72 h (Figure 8D–F). On the contrary, the percentage of erythrosine B-positive cells induced by treatment with CKs 1.5 h after TcdB (0.1 ng/mL) increased progressively up to 72 h (Figure 8D–F). In fact, we found 24.77% of erythrosine B-positive cells at 24 h, 58.30% at 48 h, and 76.10% at 72 h (Figure 8D–F).

The results of the number of live cells showed that after treatment with TcdB alone at 0.1 ng/mL they were decreased at 72 h, with a reduction of about 1.83-fold in the live cell number (Figure 8I) with respect to the number at 24 h (Figure 8G). The treatment with CKs 1.5 h after TcdB at 0.1 ng/mL strongly and progressively decreased the number of live cells until 72 h (Figure 8H,I) with respect to the treatment with TcdB alone both at the same time (Figure 8H,I) and at 24 h (Figure 8G).

Analysis of the cell cycle showed that after treatment with TcdB alone at 0.1 ng/mL there was a strong cell cycle arrest in G2/M of EGCs; indeed, we found an increase of cells in the G2/M phase with the disappearance of the S phase that start at 24 h and was maintained over time (Figure 8J–L). The treatment with CKs 1.5 h after 0.1 ng/mL did not further increase the cell cycle arrest in G2/M (Figure 8J–L). Therefore, at these times (48 h and 72 h), the cell cycle arrest in G2/M was maintained.

Therefore, cytotoxicity induced by CKs increases over time with the maximum effect observed after a prolonged CK stimulation, indicating that in the time course, cells that have not yet undergone apoptosis and necrosis after 24 h are still affected.

## 3. Discussion

In the context of gastrointestinal infections, CDI has become a serious public health problem, both due to the number of deaths and the ever-increasing healthcare costs [11,12,13,14]. The progressive endemic diffusion of *C. difficile*, the continuous appearance of more virulent strains, and the ability to mutate the molecular domains of TcdB and TcdA involved in binding to the various Tcd receptors (chondroitin sulphate proteoglycan 4; frizzled proteins 1, 2 and 7; poliovirus receptor-like protein; sucrase isomaltase; and glycoprotein 96) differently expressed on various cell types [17,18,42], together favour greater pathogenicity, highlighting the need to develop more effective strategies to combat this serious infection.

The strategies against CDI adopted so far have shown notable limitations and can be summarized as follows: (a) new, more specific drugs that have less impact on the host’s microbiota, (b) effective vaccines directed towards Tcds, and (c) faecal transplant [43,44,45,46,47,48].

The slow progress of these therapeutic strategies encourages further investigation into the pathogenicity mechanisms of Tcds to try to discover new targets on which to operate.

We have previously demonstrated that TcdB, in the presence of the pro-inflammatory CKs, increases its cytotoxicity towards EGCs [28,37], cells of paramount importance for the pathophysiological regulation of the intestinal functions [38,39,40,41]. This “cytotoxic synergism” between TcdB and CKs could represent a new mechanism adopted by *C. difficile* for enhancing cell death such as apoptosis and necrosis. In fact, we have demonstrated that the apoptotic pathways activated by TcdB alone are all enhanced by the presence of CKs [28,37].

Given the importance of the inflammatory response in containing an infection [1,2,3,4] and the well-known harmful side effects of the inflammatory response [5,6,7,8], our previous demonstration of apoptotic cytotoxic synergism between TcdB and CKs [28,37] led us to suggest that *C. difficile* may have evolved a strategy to use some cytokines produced in contest of the inflammatory response to enhance its cytotoxic ability.

In the present study, we wanted to analyse whether the timing of stimulation with pro-inflammatory CKs of cells infected with TcdB (time 0) affects cytotoxic synergism, evaluated as apoptosis and necrosis.

In the schemes chosen to evaluate the effects of cytotoxic synergism between TcdB and the timing of the appearance of CKs, attempting to simulate what could happen in vivo, we continued the observation time of the effects after 24 h because very important cytotoxic effects occur subsequently that would not have been observed if the experiments were stopped at 24 h. In particular, we have chosen the specific time intervals varying from 1.5 h to 24 h, 48 h, 72 h, and 96 h from infection because cell death (above all apoptosis but also necrosis) can occur a long time after triggering. This experimental approach made it possible to demonstrate how CKs progressively led to cell death when applied to cells after TcdB, firstly by apoptosis and then necrosis. Whereas the TcdB alone slightly increased cell death after 24 h with TcdB at 0.1 ng/mL and reduced cell death by apoptosis with TcdB at 1 ng/mL and 10 ng/mL, because at 48 h, cells make the choice between apoptosis or senescence, and at this time, stop the induction of apoptosis and start the process to undergo cellular senescence [28,37,49,50].

The various schemes all represent situations of possible interactions between *C. difficile* and the timing of CK stimulation; therefore, there is no advantage or benefit of one scheme over the other but the study of all of them is important to understand the role of the inflammatory response in the pathogenesis of CDI.

The results demonstrated that CKs strongly enhanced cell death by apoptosis but also by necrosis induced by TcdB in EGCs. The enhancement of apoptosis by CKs was dependent on the TcdB dose and in a less relevant manner on the timing of CK stimulation, while enhancement of necrosis occurred in a manner independent of the TcdB dose and the timing of CK stimulation. In fact, the most relevant result is that CKs are always able to enhance cell death if they come into contact with EGCs before, concomitantly, or shortly after an interaction of TcdB with EGCs, and also even 48 h after an interaction of TcdB with EGCs. Furthermore, the three doses of TcdB used allowed us to highlight an important phenomenon, i.e., a greater enhancement of apoptosis occurred with the lower dose of TcdB (0.1 ng/mL) in all the of conditions examined. CKs with a high dose of TcdB (10 ng/mL) significantly increase apoptosis only if they come into contact with EGCs 48 h after the interaction of TcdB with EGCs or after prolonged CKs stimulation. 

CKs, when added a longer time from TcdB intoxication, strongly enhanced the cytotoxic activity of TcdB, both as apoptosis and necrosis; indeed, the cytotoxic activity in apoptosis of TcdB+CKs with all doses of TcdB strongly increased with time despite that apoptosis induced by TcdB alone at 1 ng/mL and 10 ng/mL decreased with time, likely because, as previously demonstrated, these doses of TcdB alone lead to cellular senescence of EGCs [50]. We previously demonstrated that at 48 h, EGCs treated with TcdB alone at 10 ng/mL make the choice between apoptosis or senescence, and at this time, stop the induction of apoptosis and start the process to undergone cellular senescence [50]. However, cytotoxic synergism in apoptosis with TcdB+CKs was greater when EGCs incubated with TcdB were stimulated with CKs for 48 h (+96 h from time 0, scheme E in Figure 2). Also, the cytotoxic synergism in necrosis with all doses of TcdB was greater when EGCs infected with TcdB were stimulated with CKs for 48 h (+96 h from time 0, scheme E in Figure 2).

However, CKs strongly increased necrosis for all doses of TcdB used and time of contact with EGCs; indeed, we found an increase of erythrosine B-positive cells with all TcdB doses and times of contact with EGCs. Moreover, a further relevant fact is that when the cells that interacted with the TcdB plus CKs were observed for longer time, the cell death (apoptosis and necrosis) progressively increased over time with respect to 24 h, leading to the death of the majority of cells. In fact, those cells that after 24 h were not yet apoptotic and necrotic underwent cell death, as indicated by the fact that a strong decrease in the number of live cells was detected only at the longest time.

However, at shorter times (24 h) of CK stimulation, the more relevant cytotoxic synergistic effects were observed with the CKs and TcdB at a low dose (0.1 ng/mL), but at longer times (48–72 h) of CKs stimulation, a strong cytotoxic synergism in apoptosis and necrosis was induced by all doses of TcdB. When CKs stimulation occurred after a longer time from TcdB infection, a strong cytotoxic synergism in apoptosis and necrosis was induced by all doses of TcdB and also at shorter times of CKs stimulation, although it was the strongest at a longer time.

CKs seem to act on EGCs that have interacted with TcdB, like a rheostat, with apoptosis becoming necrosis when the dose of TcdB increases. 

These observations of strong enhancement by CKs of TcdB-induced cell death might have a possible correspondence in vivo. During infection, the various types of colon cells undergo interactions with different concentrations of TcdB. Thus, this could imply that the pro-inflammatory cytokines not only directly enhance the cytotoxicity of TcdB, but also that the cells that have interacted with low doses of TcdB, and are possibly recovering after a temporary arrest of the cell cycle, progressively undergo cell death. Therefore, most cells interacting with TcdB in the presence of an inflammatory response are destined to eventually die.

The results of cell viability assay and cell cycle analysis showed that TcdB alone caused a strong reduction of the number of live cells and cell cycle arrest in G2/M at all concentrations of TcdB used with respect to control EGCs, while the stimulation with CKs before and also, after a relatively longer time, the infection with TcdB reduce the number of live cells but did not further increase cell cycle arrest in G2/M with TcdB at 0.1 ng/mL and 1 ng/mL, and increased cell cycle arrest with TcdB at 10 ng/mL. Therefore, the reduction of the live cell number after TcdB alone is due both to cell cycle arrest and cell death from necrosis, while after the stimulation with CKs in all experimental schemes it is due only to an increase in cell death from necrosis.

What approach would be possible to antagonize this cytotoxic synergism? During the period in which a specific antibiotic therapy keeps CDI under control and progressively eliminates the bacterium, the reduction of pro-inflammatory circulating cytokines as TNF-α and IFN-γ could reduce the most severe effects of the cytotoxic synergistic cell death that TcdB is capable of causing. Of course, once the infection has been eliminated, the reappearance of low levels of pro-inflammatory CKs could still lead to death of the cells that have already interacted with TcdB; this event, however, could be less relevant compared to the massive cell death caused by the CKs during the acute phase of CDI. In fact, we have previously demonstrated that the cytotoxic synergism between TcdB and CKs is dependent on the CK dose, and that it disappears at lower doses [28].

Furthermore, assays are now available for the rapid and sensitive detection of TcdB in the faeces, which allows real-time monitoring of the presence of TcdB in the colon of infected subjects [51].

Of course, after antibiotic therapy, with CDI temporarily under control, the pro-inflammatory CK response no longer antagonized will contribute to the eradication of *C. difficile* not completely eliminated by antibiotic therapy.

Further studies are obviously needed to understand the molecular logic of this cytotoxic synergism and how much relevance has in vivo because if we adequately develop new therapeutic strategies, this cytotoxic synergism could paradoxically be the Achilles’ heel of CDI.

## 4. Materials and Methods

### 4.1. TcdB

TcdB was isolated from *C. difficile* strain VPI10463 and obtained from Enzo Life Sciences (BML-G150-0050; Farmingdale, NY, USA). To prepare a stock solution, TcdB was reconstituted to 200 µg/mL and stored at −80 °C, as indicated in the data sheet, before use in the experiments at a concentration of 0.1 ng/mL, 1 ng/mL, and 10 ng/mL [28,37,50].

### 4.2. Cell Culture and Treatment with TcdB

Rat-transformed EGCs [EGC/PK060399egfr (ATCC CRL-2690)] [52], obtained from the ATCC (Manassas, VA, USA), were cultured in Dulbecco’s modified Eagle’s medium (DMEM) (Sigma Aldrich, Milano, Italy) with 10% foetal bovine serum (Gibco/Fisher Scientific, Hampton, NH, USA), 2 mM L-glutamine, 1× penicillin plus streptomycin (all from Euroclone, Milano Italy) (complete medium) at 37 °C with 5% CO_2_ for no more than 20 passages [28,50].

EGCs were released using 0.05% trypsin–EDTA (Euroclone, Milano Italy) seeded at a density of 0.5 × 10^6^ cells/well in 2 mL of complete medium in six-well culture plates, and allowed to adhere. Thereafter, we simulated the three main situations of possible interactions between *C. difficile* and the host inflammatory response, treating the EGCs with the proinflammatory cytokines, TNF-α (#300-01A PeproTech, Rocky Hill, NJ, USA) plus IFN-γ (#400-20 PeproTech, Rocky Hill, NJ, USA), and TcdB with the following experimental schemes:

First Situation:

Scheme A: EGCs were stimulated for 18 h (−18 h) with 50 ng/mL TNF-α plus 50 ng/mL IFN-γ (CKs), then treated with TcdB at 0.1 ng/mL, 1 ng/mL, or 10 ng/mL (time 0), then were recovered at 24 h (scheme A in Figure 2).

Scheme B: EGCs were stimulated for 2 h (−2 h) with CKs, then treated with TcdB at 0.1 ng/mL, 1 ng/mL, or 10 ng/mL (time 0), and were recovered at 24 h (scheme B in Figure 2).

Second Situation:

Scheme C: EGCs were treated with TcdB at 0.1 ng/mL, 1 ng/mL, or 10 ng/mL (time 0) and simultaneously (0 h) stimulated with CKs, and were recovered at 24 h (scheme C in Figure 2).

Scheme D: EGCs treated with TcdB at 0.1 ng/mL, 1 ng/mL, or 10 ng/mL (time 0) after 1.5 h (+1.5 h) were stimulated with CKs, then were recovered at 24 h (scheme D in Figure 2).

Third Situation:

Scheme E: EGCs treated with TcdB at 0.1 ng/mL, 1 ng/mL, or 10 ng/mL (time 0) after 48 h (+48 h) were stimulated with CKs for 24 h and 48 h, then were recovered at 24 h (+72 h from time 0) or 48 h (+96 h from time 0) (scheme E in Figure 2).

Scheme F: EGCs treated with TcdB at 0.1 ng/mL, 1 ng/mL, or 10 ng/mL (time 0) after 1.5 h (+1.5 h) were stimulated with CKs, then were recovered at 24 h or 48 h (scheme F in Figure 2).

Scheme G: EGCs treated with TcdB at 0.1 ng/mL (time 0) were after 1.5 h (+1.5 h) stimulated with CKs, and recovered at 24 h, 48 h, or 72 h (scheme G in Figure 2).

Then, cells from all of the experimental conditions at the time indicated were recovered as described above, washed, and cell viability and live cell numbers were determined by an erythrosine B dye-exclusion assay.

After measurement of the viability and number of live cells, the cells were aliquoted at 0.5 × 10^6^ in 12 × 75-mm tubes and the percentage of apoptosis was evaluated by flow cytometry [28,37,50].

### 4.3. Evaluation of Cell-Cycle and Apoptosis by Flow Cytometry

The control and EGCs treated as described above were recovered at the time indicated and analysed by flow cytometry to evaluate the DNA content and detect apoptosis and cell-cycle changes [28,37,50].

For this purpose, the cell pellets were resuspended in 1 mL of a hypotonic fluorochrome solution containing 50 μg/mL propidium iodide (PI, Sigma Aldrich, Milano, Italy) in 0.1% sodium citrate and 0.1% Triton X-100 (all from Sigma Aldrich, Milano, Italy) [28,37,50,53,54]. The samples were incubated for 2 h at 4 °C in the dark, and the PI fluorescence of each nucleus was evaluated with an EPICS XL-MCL flow cytometer (Beckman Coulter, Miami, FL, USA) [28,37,50,53,54]. The data were processed with an Intercomp computer and analysed with EXPO32 ADC software (Beckman Coulter, Miami, FL, USA) [28,37,50,53,54]. Apoptosis was analysed by assessing DNA-bound PI fluorescence in the red fluorescence channel (FL3) using a logarithmic amplification, as described by Nicoletti et al. [55]. For analysis of the percentage of apoptotic cells (hypodiploid DNA content) we used the EXPO32 ADC software (Beckman Coulter, Miami, FL, USA) [28,37,50,53,54]. Flow cytometry analyses of apoptosis were repeated in six independent experiments and the data are the mean ± standard deviation of the percentage of hypodiploid nuclei obtained. The data were analysed as described in the Statistical analysis. The cell cycle phases were analysed assessing DNA-bound PI fluorescence in the orange-red fluorescence channel (FL2) using a linear amplification. For analysis of the percentage of cells in each cell-cycle phase, we used ModFit 1.0 software (Verity Software House, Topsham, ME, USA) [28,37,50]. Flow cytometry analyses of the cell cycle were repeated in six independent experiments and the data are the mean ± standard deviation of the percentage of cells in each cell-cycle phase. The data were analysed as described in the Statistical analysis.

### 4.4. Statistical Analysis

All data are expressed as mean ± standard deviation. GraphPad Prism 9.0.0 software was used to make statistical charts. The Shapiro–Wilk normality test was applied to analyse the data normality distribution. Comparisons of multiple groups were analysed using one-way ANOVA followed by Tukey’s post hoc test for normally distributed data for evaluating the significance of the differences between the groups. *p*-values = or less than 0.05 were defined as statistically significant.

## 5. Conclusions

Pro-inflammatory CKs increase the cytotoxicity of TcdB on EGCs if stimulation occurs over a period of time, from −18 h to +48 h relative to the time of TcdB infection (0 h). Furthermore, cell death induced by TcdB+CKs switches from apoptosis to necrosis, which progressively involves the majority of cells that come into contact with TcdB. This cytotoxic synergism, which occurs in all experimental conditions tested, when confirmed in other experimental models, can become relevant as a new pathogenetic mechanism of TcdB with the possibility of antagonizing it in a therapeutic function.

## Figures and Tables

**Figure 1 ijms-25-00958-f001:**
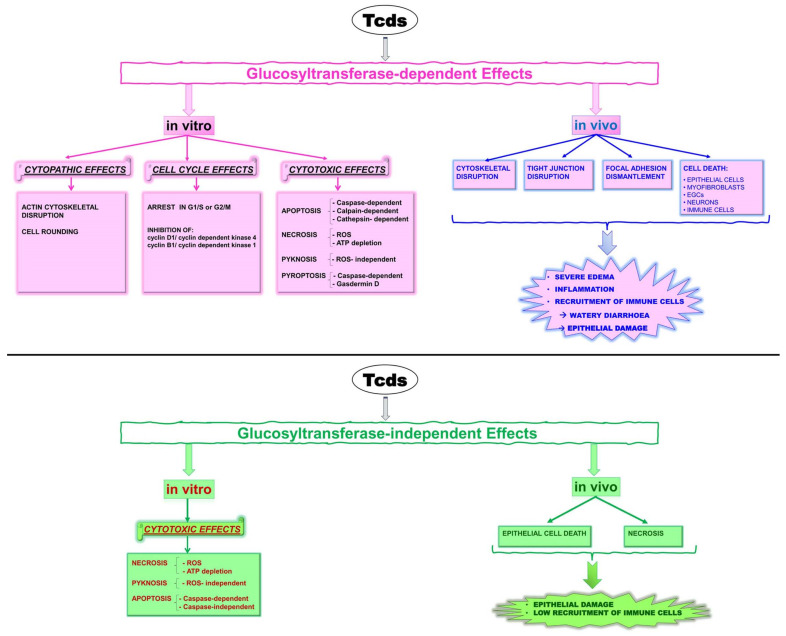
Effects of Tcds in vitro and in vivo, both glucosyltransferase-inhibition-dependent and glucosyltransferase-independent. Abbreviations: *Clostridioides difficile* toxins (Tcds), reactive oxygen species (ROS), enteric glial cells (EGCs).

**Figure 2 ijms-25-00958-f002:**
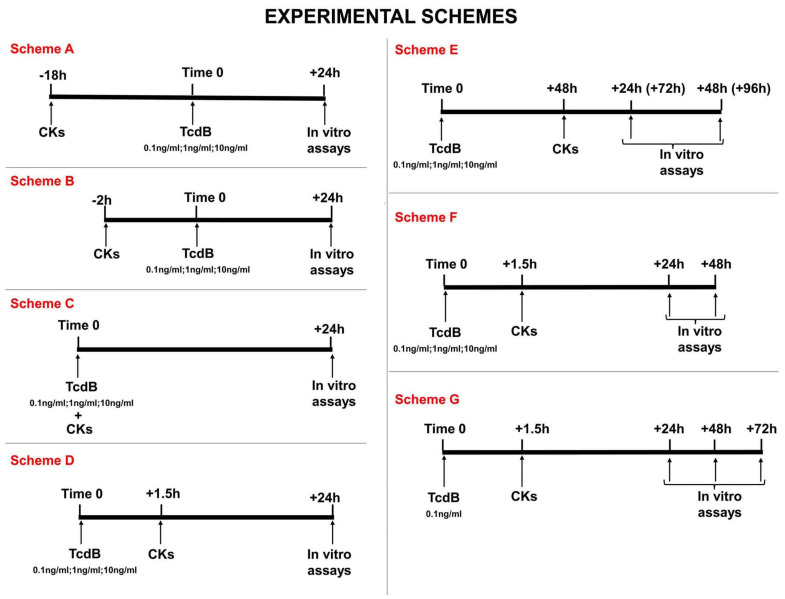
Experimental schemes for simulation of the three main situations of possible interactions between *C. difficile* TcdB and the host inflammatory response. In (**A**–**G**) we schematized the in vitro treatment in our experimental model with TcdB and EGCs for simulation of the three main situations of possible interactions between *C. difficile* toxin B and the timing of CKs stimulation. Abbreviations: tumor necrosis factor alpha (TNF-α); interferon gamma (IFN-γ); pro-inflammatory cytokines TNF-α plus IFN-γ (CKs); *Clostridiodes difficile* toxin B (TcdB).

**Figure 3 ijms-25-00958-f003:**
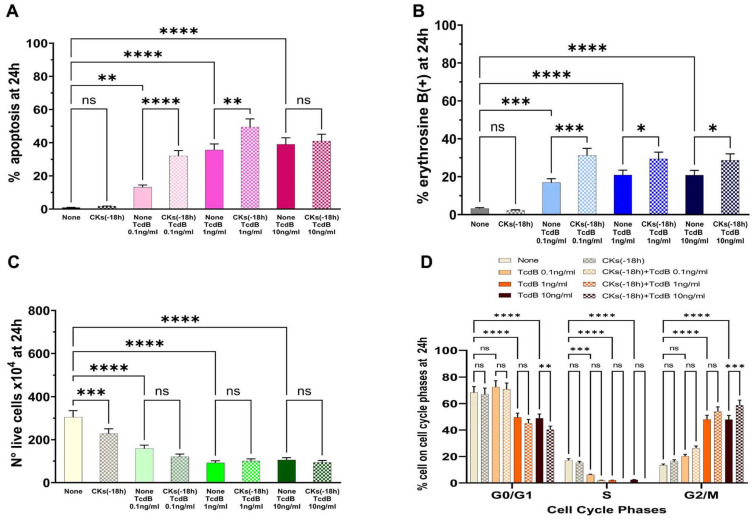
Effect of CKs added 18 h before TcdB infection on EGC apoptosis, cell viability, cell growth, and cell cycle phases. EGCs were pre-treated or not for 18 h (−18 h) with 50 ng/mL TNF-α plus 50 ng/mL IFN-γ (CKs), then treated with TcdB at 0.1 ng/mL, 1 ng/mL, or 10 ng/mL (time 0), and the cells were recovered at 24 h (scheme A in Figure 2). At the time indicated was determined: (**A**) apoptosis by measuring the percentage of hypodiploid nuclei by flow cytometry; the percentage of erythrosine B(+) cells (**B**) and the live cell number (**C**) by the erythrosine B dye-exclusion assay; and (**D**) the percentage of cells in each cell-cycle phase by flow cytometry with ModFit 1.0 software. (**A**–**D**) The data are the mean ± standard deviation of six experiments. Statistical analysis was performed by one-way ANOVA and Tukey’s multiple comparisons test. * *p* < 0.05, ** *p* < 0.01, *** *p* < 0.001, **** *p* < 0.0001, not significant (ns) *p* > 0.05.

**Figure 4 ijms-25-00958-f004:**
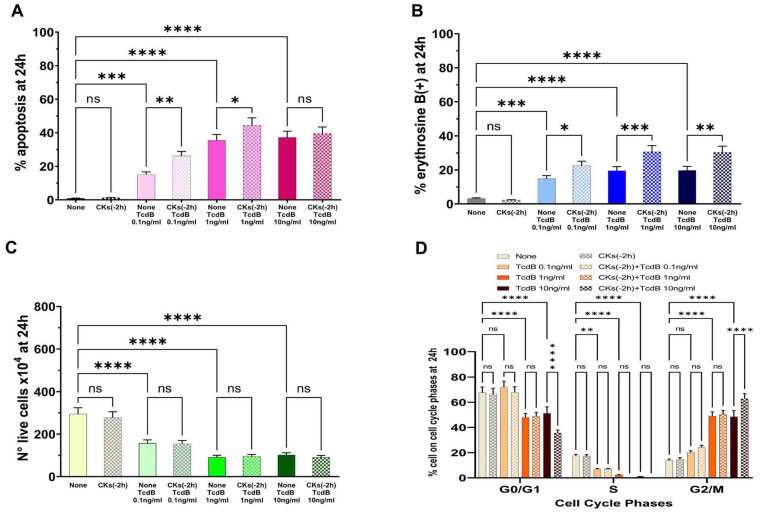
Effect of CKs added 2 h before TcdB infection on EGC apoptosis, cell viability, cell growth, and cell cycle phases. EGCs were pre-treated or not for 2 h (−2 h) with CKs, then treated with TcdB at 0.1 ng/mL, 1 ng/mL, or 10 ng/mL (time 0), and then the cells were recovered at 24 h (scheme B in Figure 2). At the time indicated was determined: (**A**) apoptosis by measuring the percentage of hypodiploid nuclei by flow cytometry; the percentage of erythrosine B(+) cells (**B**) and the live cell number (**C**) by the erythrosine B dye-exclusion assay; and (**D**) the percentage of cells in each cell-cycle phase by flow cytometry with ModFit 1.0 software. (**A**–**D**) The data are the mean ± standard deviation of six experiments. Statistical analysis was performed by one-way ANOVA and Tukey’s multiple comparisons test. * *p* < 0.05, ** *p* < 0.01, *** *p* < 0.001, **** *p* < 0.0001, ns *p* > 0.05.

**Figure 5 ijms-25-00958-f005:**
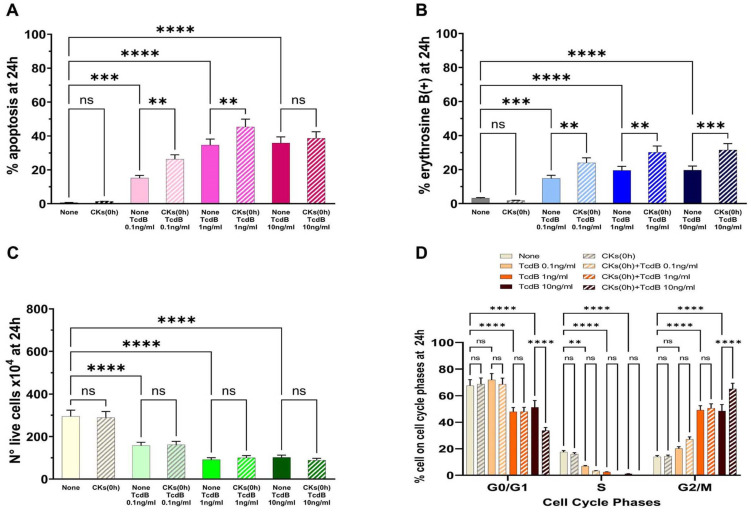
Effect of CKs added concomitantly with TcdB infection on EGC apoptosis, cell viability, cell growth, and cell cycle phases. EGCs were treated with TcdB at 0.1 ng/mL, 1 ng/mL, or 10 ng/mL (time 0) and simultaneously (0 h) stimulated with CKs, then the cells were recovered after 24 h (scheme C in Figure 2). At this time was determined: (**A**) apoptosis by measuring the percentage of hypodiploid nuclei by flow cytometry; the percentage of erythrosine B(+) cells (**B**) and the live cell number (**C**) by the erythrosine B dye-exclusion assay; and (**D**) the percentage of cells in each cell-cycle phase by flow cytometry with ModFit 1.0 software. (**A**–**D**) The data are the mean ± standard deviation of six experiments. Statistical analysis was performed by one-way ANOVA and Tukey’s multiple comparisons test. ** *p* < 0.01, *** *p* < 0.001, **** *p* < 0.0001, ns *p* > 0.05.

**Figure 6 ijms-25-00958-f006:**
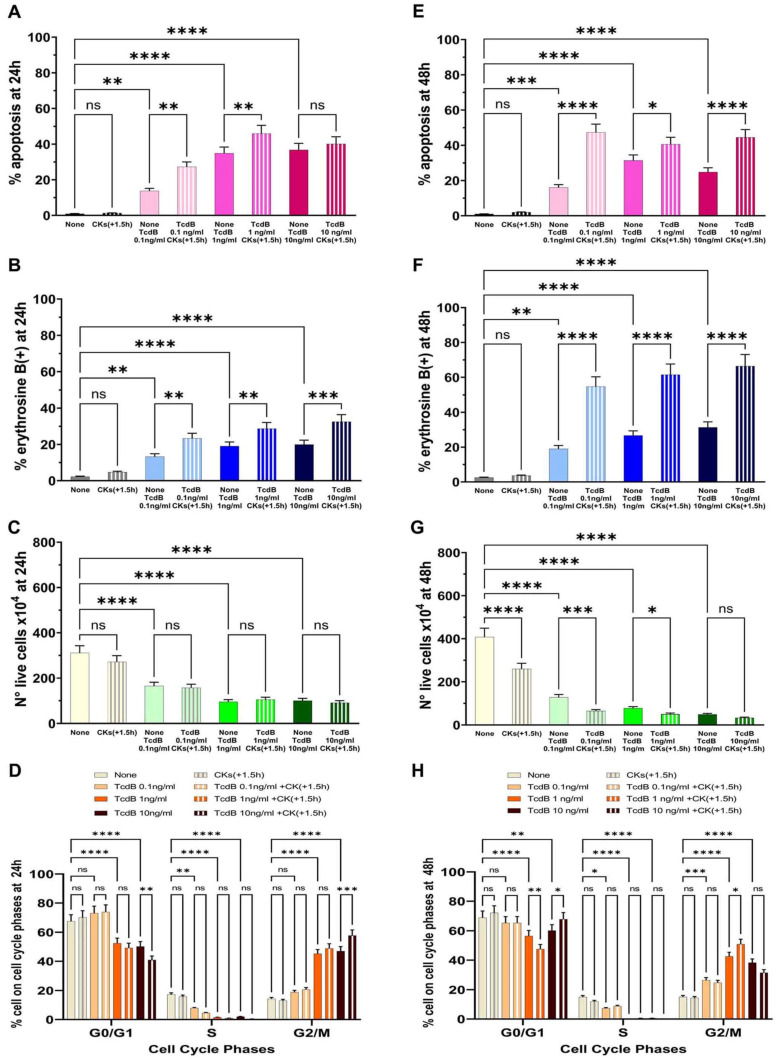
Effect of CKs added 1.5 h after TcdB infection on EGC apoptosis, cell viability, cell growth, and cell cycle phases. EGCs treated with TcdB at 0.1 ng/mL, 1 ng/mL, or 10 ng/mL (time 0) were after 1.5 h (+1.5 h) stimulated with CKs, and then the cells were recovered at 24 h or 48 h (scheme D,F in Figure 2). At these times was determined: (**A**,**E**) apoptosis by measuring the percentage of hypodiploid nuclei by flow cytometry; the percentage of erythrosine B(+) cells (**B**,**F**) and the live cell number (**C**,**G**) by erythrosine B dye-exclusion assay; and (**D**,**H**) the percentage of cells in each cell-cycle phase by flow cytometry with ModFit 1.0 software. (**A**–**H**) The data are the mean ± standard deviation of six experiments. Statistical analysis was performed by one-way ANOVA and Tukey’s multiple comparisons test. * *p* < 0.05, ** *p* < 0.01, *** *p* < 0.001, **** *p* < 0.0001, ns *p* > 0.05.

**Figure 7 ijms-25-00958-f007:**
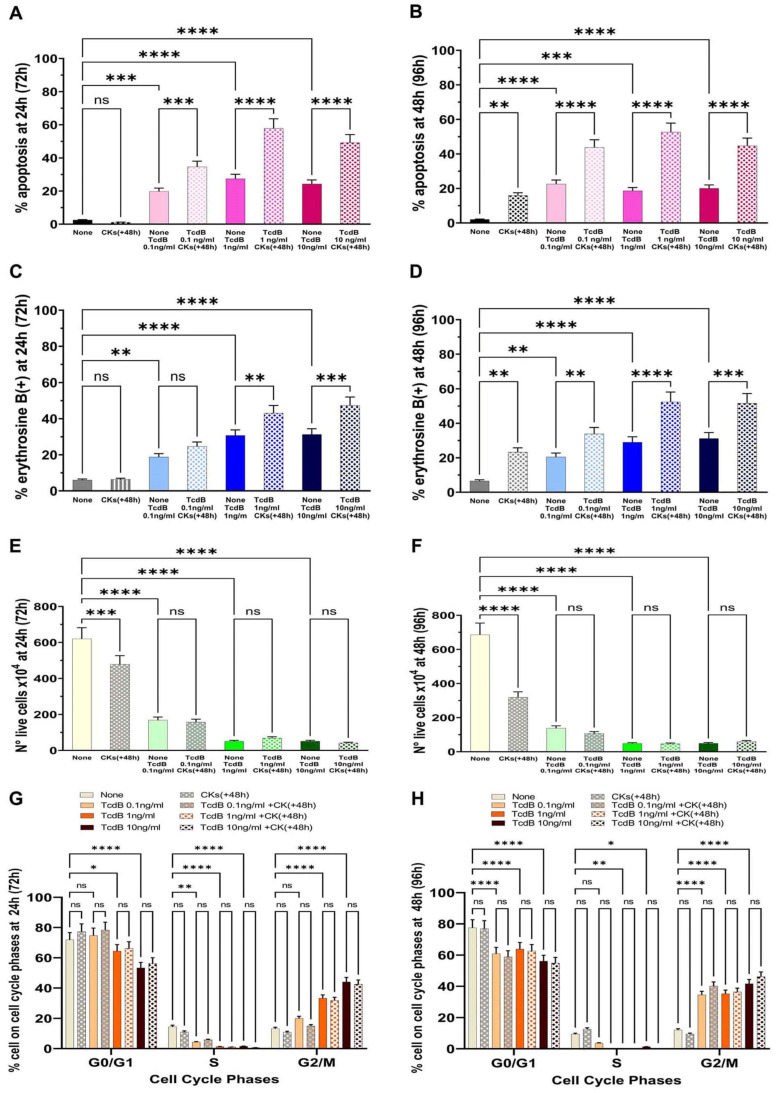
Kinetics of the effect of CKs added 48 h after TcdB infection on EGC apoptosis, cell viability, cell growth, and cell cycle phases. EGCs treated for 48 h with TcdB at 0.1 ng/mL, 1 ng/mL, or 10 ng/mL (time 0) were stimulated with CKs for 24 h or 48 h and the cells were recovered at 24 h (+72 h from time 0) or 48 h (+96 h from time 0) (scheme E in Figure 2). At these times was determined: (**A**,**B**) apoptosis by measuring the percentage of hypodiploid nuclei by flow cytometry; the percentage of erythrosine B(+) cells (**C**,**D**) and the live cell number (**E**,**F**) by erythrosine B dye-exclusion assay; and (**G**,**H**) the percentage of cells in each of the cell-cycle phases by flow cytometry with ModFit 1.0 software. (**A**–**H**) The data are the mean ± standard deviation of six experiments. Statistical analysis was performed by one-way ANOVA and Tukey’s multiple comparisons test. * *p* < 0.05, ** *p* < 0.01, *** *p* < 0.001, **** *p* < 0.0001, ns *p* > 0.05.

**Figure 8 ijms-25-00958-f008:**
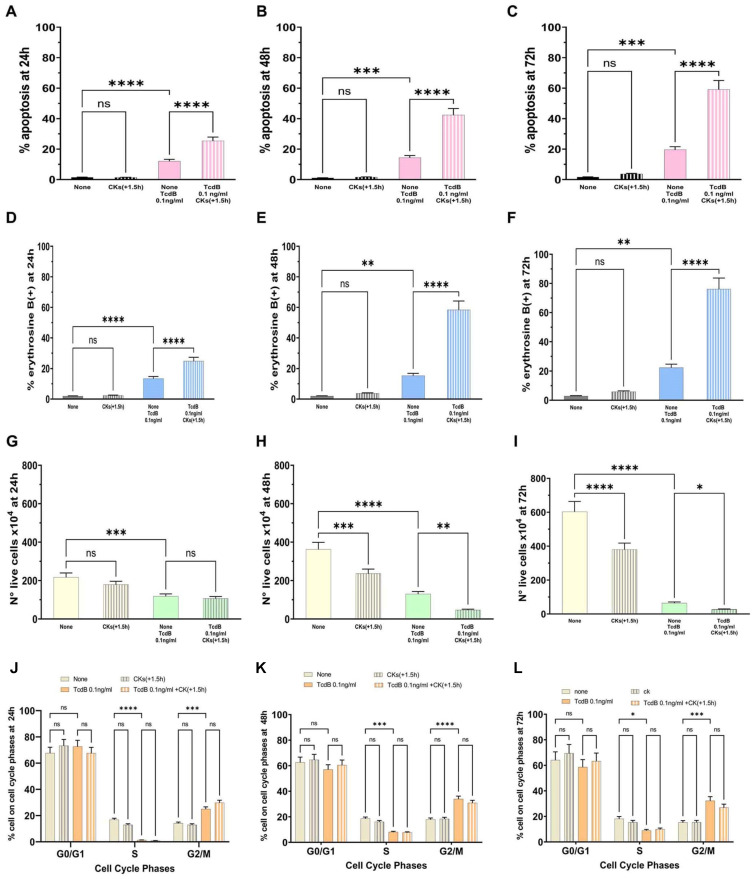
Kinetics of the effects of CKs added 1.5 h after TcdB at 0.1 ng/mL on EGC apoptosis, cell viability, cell growth, and cell cycle phases. EGCs treated for 1.5 h with TcdB at 0.1 ng/mL (time 0) were stimulated with CKs and cells were recovered at 24 h, 48 h, and 72 h (scheme G in Figure 2). At these times was determined: (**A**–**C**) apoptosis by measuring the percentage of hypodiploid nuclei by flow cytometry; the percentage of erythrosine B(+) cells (**D**–**F**) and the live cell number (**G**–**I**) by erythrosine B dye-exclusion assay; and (**J**–**L**) the percentage of cells in each cell-cycle phase by flow cytometry with ModFit 1.0 software. (**A**–**L**) The data are the mean ± standard deviation of six experiments. Statistical analysis was performed by one-way ANOVA and Tukey’s multiple comparisons test. * *p* < 0.05, ** *p* < 0.01, *** *p* < 0.001, **** *p* < 0.0001, ns *p* > 0.05.

## Data Availability

Data are contained within the article.

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
