# Peer review of "Pro-Inflammatory Cytokines Enhanced In Vitro Cytotoxic Activity of Clostridioides difficile Toxin B in Enteric Glial Cells: The Achilles Heel of Clostridioides difficile Infection?"

_ijms, 2024, doi:10.3390/ijms25020958_

Round 1

Reviewer 1 Report

Comments and Suggestions for Authors

This manuscript submitted by Katia Fettucciari  et al presents a “Pro-Inflammatory Cytokines Enhanced In vitro Cytotoxic Activity of Clostridioides difficile Toxin B in Enteric Glial Cells”

The manuscript presents lots of research data but too much data even makes a hard to follow the story. And story's conclusion was not mentioned, so it is difficult to catch up with the author's suggestions point. The current manuscript needs to improve the data presentation and the sentence for the general reader.

First, what is the benefit between Scheme A to Scheme G? What is the best readout?

In Figure 2: How about the static difference between TcdB 0.1 to 10 ng/ml with CKs?

“erythrosine B + cells” change to “erythrosine B(+) cells: 

What is the difference between the number of live cells and the number of total cells?

it should be the same trend. If the author wants to get a claim for cell death data, it should show cell death, not total cells.

In Figure 3: why not increase the cell death with TcdB at 10 ng/ml and compare with TcdB at 0.1 ng/ml?

Line 251: typo of Figure 4A.

Line 278: what does “loss of cells for lysis”?

In Figure 5: How can it be less apoptosis occurred at 48 h compared to 24 h?

Line 414: change to 24.77%.

“erythrosine B + cells” change to “erythrosine B(+) cells.

Some of the references are missing page numbers.

Comments on the Quality of English Language

the same meaning of words and sentences were repeated. 

Reviewer 2 Report

Comments and Suggestions for Authors

The manuscript presents an experimental study evaluating the role of C. difficile and the effect of Cdiff toxins on the pro-inflammatory cytokines stimulation. The topic is of interest to researchers focusing on the preclinical studies on this topic, offering potential insight into the pathophysiology of Cdiff infection.

There are also several aspects to be considered:

-        The introduction section should be shortened and should end with clearly stating the aim of the study, without description of methodology in the introduction.

-        The authors should include a figure synthesizing both de Glucosyltransferase-dependent activity of Tcds and the glucosyltransferase-independent effects, as a visual aid to enhance interest of the reader.

-        The authors should include the methods section after the introduction and to include explanations on the choice of statistical tests; moreover, they should remove part of methods related data described in the introduction

-        The authors should explain why they chose those specific time intervals from infection to report the data, varying from 1.5h to 24…72h, 96h.

-        From my point of view, the authors should rearrange the way results are reported, since data is very difficult to follow and crowded.

-        For most of the data, authors mention “The data are the mean ± standard deviation of six experiments”, however, no information on whether the values were normally distributed is available in the results section. Therefore, the results section should begin with presenting some descriptive statistics data

-        The authors should avoid commenting on results in the results section and include this in the discussion section (eg. “In the light of these results of cell cycle analysis the strong reduction of the number 276 both of live cells and total cells at all concentrations of TcdB used with respect to control 277 EGCs was due both to cell cycle arrest and loss of cells for lysis.”)

-        Some statements should be rephrased, such as “Altogether, these results confirm that cytotoxicity increases over time and highlight that cells that meet TcdB and undergo a prolonged stimulation with CKs are undergoing a strong increase in cytotoxicity both by apoptosis and necrosis in cells which after 24h are not yet apoptotic and necrotic.” and cut in shorter sentences to facilitate the clarity of phrasing. Moreover, this phrase would better fit the discussion section.

-        Since the discussion section is large, the authors should also add a conclusions section to clearly shape the conclusions related to this experimental study.

Comments on the Quality of English Language

Minor editing of English language is required, but phrasing in shorter and clearer sentences is necessary.
